

# Sorghum and wheat differentially affect caecal microbiota and associated performance characteristics of meat chickens

Eduardo Crisol-Martínez[1,*], Dragana Stanley[1,2,3,*], Mark S. Geier[4], Robert J. Hughes[3,5,6] and Robert J. Moore[3,7,8]

[1] School of Medical and Applied Sciences, Central Queensland University, Rockhampton, Queensland, Australia
[2] Institute for Future Farming Systems, Central Queensland University, Rockhampton, Queensland, Australia
[3] Poultry Cooperative Research Centre, University of New England, Armidale, New South Wales, Australia
[4] Research and Innovation Services, University of South Australia, Mawson Lakes, South Australia, Australia
[5] Pig and Poultry Production Institute, South Australian Research and Development Institute, Roseworthy, South Australia, Australia
[6] School of Animal and Veterinary Sciences, The University of Adelaide, Roseworthy, South Australia, Australia
[7] School of Science, RMIT University, Bundoora, Victoria, Australia
[8] Department of Microbiology, Monash University, Clayton, Victoria, Australia
[*] These authors contributed equally to this work.

Corresponding author
Eduardo Crisol-Martínez,
eduardocrisol@gmail.com

## ABSTRACT

This study compared the effects of wheat- and sorghum-based diets on broiler chickens. The growth performance and caecal microbial community of chickens were measured and correlations between productivity and specific gut microbes were observed. Cobb broilers 15 days of age were individually caged and two dietary treatments were used, one with a wheat-based diet ($n = 48$) and another one with a sorghum-based diet ($n = 48$). Growth performance measurements were taken over a 10 day period and samples for microbiota analysis were taken at the end of that period. Caecal microbiota was characterised by sequencing of 16S bacterial rRNA gene amplicons. Overall, the results indicated that a sorghum-based diet produced higher apparent metabolisable energy (AME) and body-weight gain (BWG) values in chickens, compared to a wheat-based diet. Nevertheless, sorghum-fed birds had higher feed conversion ratio (FCR) values than wheat-fed birds, possibly because of some anti-nutritional factors in sorghum. Further analyses showed that caecal microbial community was significantly associated with AME values, but microbiota composition differed between dietary treatments. A number of bacteria were individually correlated with growth performance measurements. Numerous OTUs assigned to strains of *Lactobacillus crispatus* and Lachnospiraceae, which were prevalent in sorghum-fed chickens, were correlated with high AME and BWG values, respectively. Additionally, a number of OTUs assigned to Clostridiales that were prevalent in wheat-fed chickens were correlated with low FCR values. Overall, these results suggest that between-diet variations in growth performance were partly associated with changes in the caecal microbiota.

## INTRODUCTION

Wheat, maize and sorghum are the three most commonly used grains in the poultry industry worldwide (*Liu et al., 2014*). These cereal grains usually constitute a major proportion (60–70%) of the diet fed to broilers (*Black et al., 2005*). Although cereals represent a major source of energy for birds, there are wide variations in the energy and nutrient content of different species and cultivars of cereals (*Choct & Hughes, 1999*). Specifically, wheat is perceived as having a high feeding value for poultry, yet, in reality, some studies have indicated that there are extreme variations in the apparent metabolisable energy (AME) values of wheat-based diets in chickens (*Choct, Hughes & Annison (1999)* and references within). Sorghum, the other major cereal, presents a number of economic advantages over wheat, since it can be grown under drier conditions, and is usually less costly (*Liu, Selle & Cowieson, 2013*). Also, in comparison to wheat, sorghum is thought to have a more consistent, higher energy density (*Selle et al., 2010*).

It is recognised that diet is the major determinant of the composition of the gastro-intestinal microbiota (*Shakouri et al., 2009*). A number of components in the diet, such as content of proteins, fats and carbohydrates, can influence the composition of the gut microbiota (*Hübener, Vahjen & Simon, 2002*; *Rehman et al., 2007*). There is a symbiotic relationship between gut microbiota and the host. The complex community of microorganisms, dominated by bacteria, regulates processes related to host metabolism and immunity (*Round & Mazmanian, 2009*; *Tremaroli & Bäckhed, 2012*). Thus, it is of interest to the poultry industry to understand the influence of the gut microbiota on the host's nutrient and energy use efficiency. For instance, *Stanley et al. (2013)* found an association between caecal microbiota composition and productivity in chickens, showing that the abundance of a number of caecal bacteria were correlated with high growth performance values. Thus, a better understanding of the beneficial or detrimental role of intestinal bacteria and how to influence their abundance could result in improvements in broiler performance.

Few studies have analysed the impact of different diets on the structure of the gut microbial community. Additionally, there is limited information on the influence of cereal-based diets on specific microbial populations that could be associated with the health and productivity of the birds (*Shakouri et al., 2009*). For instance, *Torok et al. (2011)* found a number of OTUs within the caecum and the ileum which were negatively or positively correlated with feed conversion ratio (FCR) values. *Lunedo et al. (2014)* found increased enterobacteria abundance in the ileum, consistent with the low feed conversion performance of sorghum-fed chickens. In a recent study, our group identified a number of caecal bacteria associated with desirable productivity outcomes on chickens fed with the same diet, and suggested that further investigation could lead to their application as probiotics to improve bird performance (*Stanley et al., 2016*). In this study, we aimed to expand this information by analysing changes in broiler performance and caecal bacteria across wheat- or sorghum-based diets, and identifying which specific bacteria were associated with parameters of productivity in each dietary treatment.
## MATERIALS AND METHODS

### Animal ethics

The Animal Ethics Committees of the University of Adelaide (Approval No.S-2011-218) and the Department of Primary Industries and Resources, South Australia (Approval No. 25/11) approved this study. All animal work was conducted in accordance with the national and international guidelines for animal welfare.

### Animal trials

Newly-hatched male Cobb 500 broiler chickens from the Baiada Hatchery, Willaston, SA, Australia, were transferred to two floor pens (one for each experimental diet) with wood sawdust and shavings bedding material in an environmentally controlled experimental animal facility. Feed and water were supplied *ad libitum* throughout the experiment. Two commercially prepared starter diets were used in this study. One was based on wheat (500 g/kg) as the main cereal component. The other diet was formulated to the same nutrient specifications but contained sorghum (300 g/k) and wheat (300 g/kg). Each diet provided 230 g/kg crude protein and 12.55 MJ/kg as fed, and both contained monensin (65 ppm), nicarbazin (45 ppm) and zinc bacitracin (75 ppm). Both diets also contained commercial exogenous phytase and xylanase enzyme products. The diets were fed from day of hatch (day 0) until the end of the experiment on day 25.

On day 13, chicks ($n = 48$ for each dietary treatment) were transferred in pairs to 48 metabolism cages in a temperature controlled room ($23-25\ °C$). Initial placing in metabolic cages in pairs was done to minimise stress and allow the birds to adjust to cages. At day 15, birds were moved into individual cages. Individual caging allowed the precise assessment of individual feed intake, energy in feed, and unused energy remaining in faeces. The experimental design eliminated competition for feed and reduced behavioural issues affecting feed intake. Single bird caging and individual measurements and sampling were implemented in order to allow direct correlation of microbiota structure and productivity measurements on a bird by bird basis. Birds were euthanised and necropsied on day 25 and caecal contents were collected from each bird and immediately transferred to a $-20\ °C$ freezer. Samples from all birds were analysed.

FCR was calculated as a ratio of feed eaten and weight gained. Thus birds with low FCR, that needed less feed per kg gained, were the most efficient in converting feed to mass. Gross energy (GE) was measured in feed and in faeces of each individual bird using a Parr isoperibol bomb calorimeter (Parr Instrument Company, Moline, IL). Apparent metabolisable energy (AME) in MJ/kg dry matter, was calculated as ($AME_{diet} = [(GE_{diet} \times$ feed eaten$) \times (GE_{excreta} \times$ dry excreta$)]/$feed eaten/dry diet content). Body weight gain (BWG) was calculated as [weight gain (g)/start weight (g)] and feed eaten (FE) was total amount of feed eaten during the 10 day measurement time period. All of the above measurements were taken from day 15 to day 25, during the time when single birds were housed in metabolic cages.

## DNA preparation, PCR amplification of 16S rRNA gene sequences, and bioinformatic analysis

DNA was isolated using the method of *Yu & Morrison (2004)*; briefly, raw DNA extracts were obtained by initial lysis of caecal content in lysis buffer at 70 °C, followed by bead beating using a Precellys 24 instrument (Stretton Scientific Limited). This was followed by enzyme removal of RNA and protein and subsequent purification using Qiagen columns as per the supplier's instruction. The V1-V3 region of the 16S rRNA gene was amplified (forward primer (*Lane, 1991*), 5′ AGAGTTTGATCCTGG 3′; reverse primer W31 (*Snell-Castro et al., 2005*), 5′ TTACCGCGGCTGCT 3′) as detailed by *Stanley et al. (2012)*. Pyrosequencing was performed using a Roche/454 FLX+ instrument and Titanium chemistry kits according to the manufacturer's instructions. Sff file processing was done using PyroBayes (*Quinlan et al., 2008*) and inspected for chimeric sequences using Pintail (*Ashelford et al., 2005*) and error corrected using Acacia (*Bragg et al., 2012*). Further trimming was done in QIIME (*Caporaso et al., 2010*) with sequence length 300–600 bases, no ambiguous sequences, minimum average quality score of 25 and maximum of 6 bases in homopolymer runs. OTU picking was done using Uclust (*Edgar, 2010*). Taxonomy was assigned using Blast against the GreenGenes database (*DeSantis et al., 2006*) and QIIME v.1.8 defaults. Additional taxonomic assignments for OTUs of interest was performed using EzTaxon database (*Chun et al., 2007*). The complete dataset is available on MG-RAST database (http://metagenomics.anl.gov/) under library ID mgl538406.

## Statistical analysis

Permutational analysis of variance (PERMANOVA) (*Anderson, 2001*) was used to analyse differences between dietary treatments on each of the four performance measures (AME, FCR, BWG and FE) and also for differences in microbiota composition. The distance matrices were based on Euclidean distance (growth performance data) and both Weighted and Unweighted Unifrac distances (microbiota abundance data) (*Lozupone & Knight, 2005*). The PERMDISP routine was used to test for the homogeneity of dispersions (based on mean distance to group centroids), to ensure that dispersions were constant among groups (*Anderson, 2006*). Principal Coordinates Analysis (PCO) was used to visualise between-groups differences in microbiota composition, based on both Unifrac distances. Analysis of variance (ANOVA) was used to test for differences in the diversity of microbial community between dietary treatments using four diversity measures (i.e., evenness, richness, Shannon's and Inverted Simpson's indices). ANOVA was also used to test for differences in the abundance of individual microbial taxa between dietary treatments. Redundancy discrimination analysis (RDA) was used to analyse individual associations between microbiota community composition and each of the performance measures in both dietary treatments. Pearson's correlation tests were conducted to analyse the correlations between performance indicators. Spearman's correlation test was used to analyse correlations between individual OTUs and each of the performance indicators. For all ANOVA, RDA and correlation analyses, microbiota abundance data was normalised and square root transformed to reduce variance heterogeneity and increase predictive power. PERMANOVA, PERMDISP and PCO analyses were carried out using PRIMER software

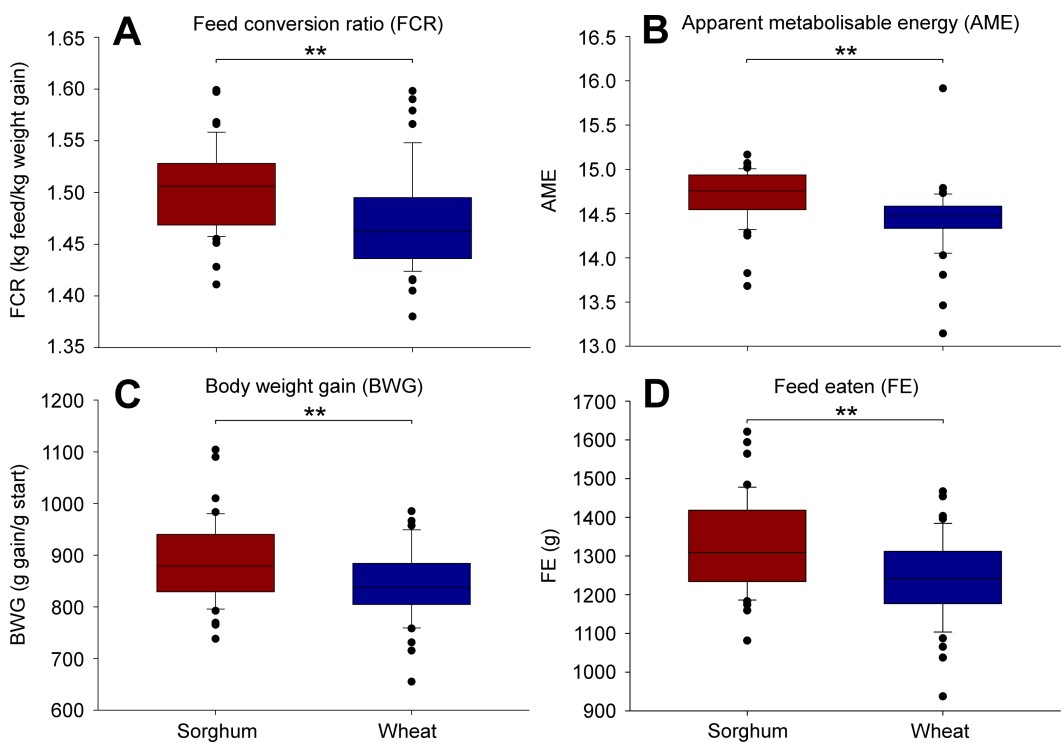

**Figure 1  Boxplots showing differences in four growth performance measures between diet treatments.** 'Sorghum' refers to birds fed with a sorghum-based diet, whereas 'wheat' refers to those fed with a wheat-based diet. Significant differences between treatments are indicated ('**': $P \leq 0.01$).

(v.6.1.16) with the PERMANOVA+add-on (v.1.0.6). RDA, Pearson's and Spearman's correlation tests, and analyses of community diversity and taxonomic structure were performed in Calypso (v.5.8) (*Zakrzewski et al., 2016*).

## RESULTS

### Growth performance measures

Dietary treatment caused significant differences in all four performance measures (Fig. 1). Chickens fed with sorghum had significantly higher FCR values (1.50 ± 0.01 [average ± standard error]) than those fed with wheat (1.47 ± 0.01) ($P = 0.004$) (Fig. 1A). Additionally, AME values were significantly higher in sorghum-fed chickens (14.71 ± 0.04) than in wheat-fed birds (14.44 ± 0.06) ($P = 0.002$) (Fig. 1B). BWG values were also significantly higher in birds fed with sorghum (886 ± 12) than in those fed with wheat (843 ± 10) ($P = 0.004$) (Fig. 1C). Lastly, chickens had significantly higher FE values when fed with sorghum (1331 ± 17) than when fed with wheat (1240 ± 16) ($P = 0.002$) (Fig. 1D). PERMDISP results showed that none of the latter PERMANOVA results were significantly affected by data dispersion (all with $P > 0.05$).

Pearson's correlation test results indicated that in each dietary treatment, AME was significantly negatively correlated with BWG (wheat: r = −0.382, $P = 0.010$; sorghum: $r = −0.410$, $P = 0.005$) and FE (wheat: r = −0.408, $P = 0.005$; sorghum: $r = −0.506$,

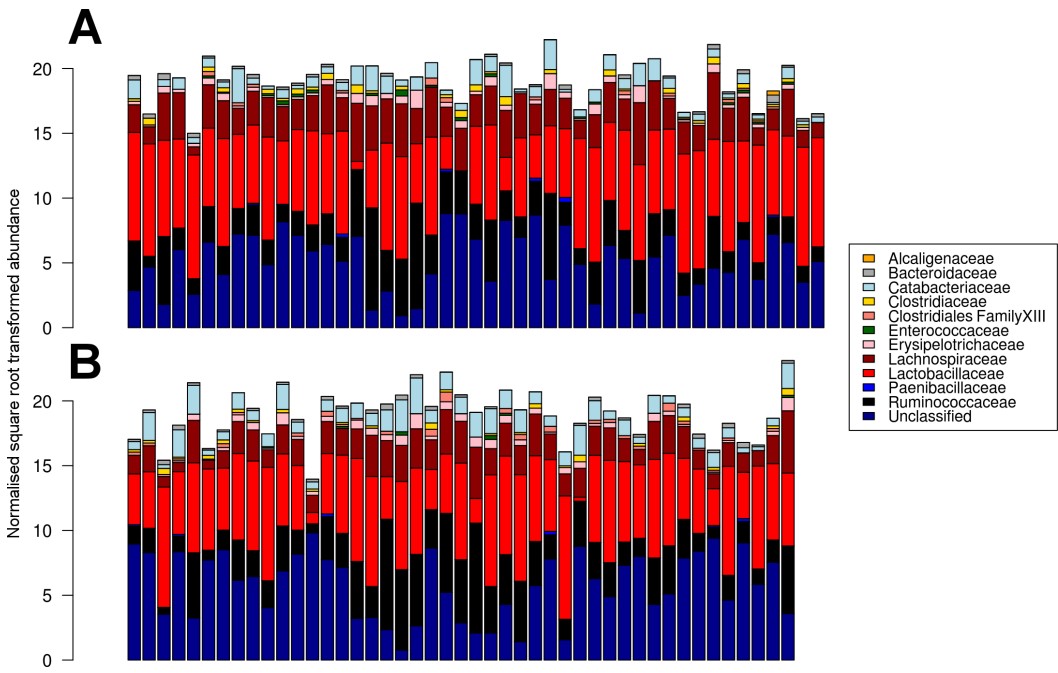

**Figure 2** Caecal microbiota profiles showing normalised square root transformed abundances of the families in sorghum-fed chickens (A) and wheat-fed chickens (B).

$P \leq 0.001$). However, FCR and FE were significantly positively correlated in wheat-fed chickens ($r = 0.311$, $P = 0.037$), but not in sorghum-fed chickens ($r = 0.155$, $P = 0.308$).

## Structure of the caecal microbial community

The microbiota profile comparison at the family level indicated high similarities between dietary treatments, both dominated by Lactobacillaceae, Lachnospiraceae and Ruminococcaceae (Fig. 2).

None of the four selected diversity measures (i.e., richness, evenness, Shannon's and inverted Simpson's indices) varied significantly between dietary treatments, neither at the OTU level, nor at any higher taxonomic level ($P > 0.05$). Comparisons of the microbiota composition at the OTU level revealed significant differences between treatments when Unweighted Unifrac Distances were used (Pseudo-$F = 2.65$, $P = 0.006$) (Fig. 3A), but not when Weighted Unifrac Distances were used (Pseudo-$F = 2.59$, $P = 0.063$) (Fig. 3B). PERMDISP confirmed that the significant result observed using Unweighted Unifrac Distances did not occur due to differences in data dispersion ($P = 0.640$).

## Gut microbiota shifts induced by diet

Diet differences produced significant changes ($P \leq 0.05$) in 98 out of 522 OTUs (18.8%). At the species level, sorghum-fed birds showed significantly higher abundances of a phylotype closest to *Lactobacillus crispatus* (identity assigned to 21 OTUs ($n = 21$), all DSM 20584(T) strains, with an averaged similarity (avg. sim.) of 92.67% [EzTaxon]) than wheat-fed birds ($P \leq 0.001$) (Fig. 4A). Out of the 32 OTUs allocated to this species, OTU3

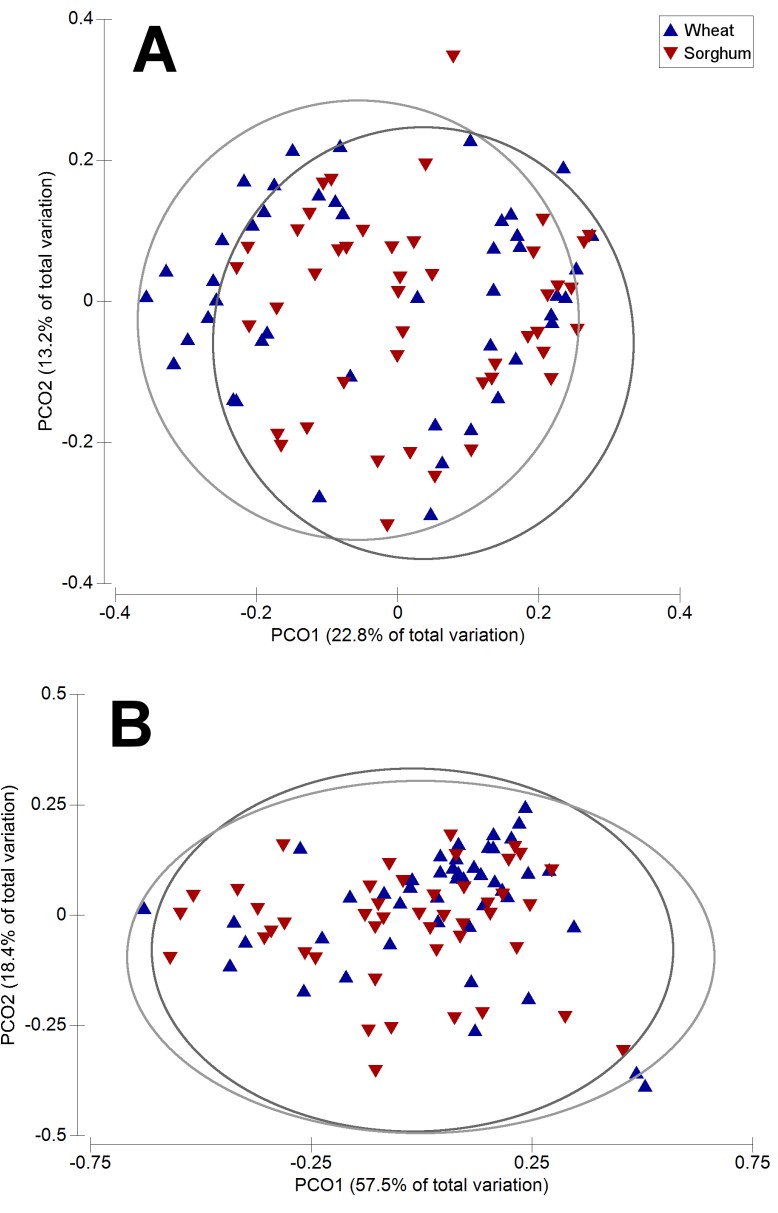

**Figure 3** **Principal Coordinates Analysis (PCO) showing a two-dimensional ordination of gut microbiota composition at the OTU level by dietary treatment (indicated by symbols).** '(A)' plot is based on Unweighted Unifrac distance, whereas '(B)' plot is based on Weighted Unifrac distance. Overlayed grey ellipses in were created to help visualise the between-treatment differences in composition. Axes indicate the percentage of variation in the data.

(with a similarity of 100%) was the most abundant and showed a similar trend as seen for the whole *L. crispatus* phylotype ($P \leq 0.001$) (Fig. 4B). Showing an opposite trend, the numbers of a phylotypes most similar to *Clostridium leptum* ($n = 30$, all DSM753 (T), avg. sim. $= 87.94\%$) were significantly higher in chickens fed with wheat than in those fed with sorghum ($P = 0.022$) (Fig. 4A). At the genus level, wheat-fed birds showed significant lower abundances of *Enterococcus* and *Coprococcus* than sorghum-fed birds ($P = 0.016$ and $P = 0.018$, respectively) (Fig. 4C). Shifts in the *Enterococcus* genus (Fig.

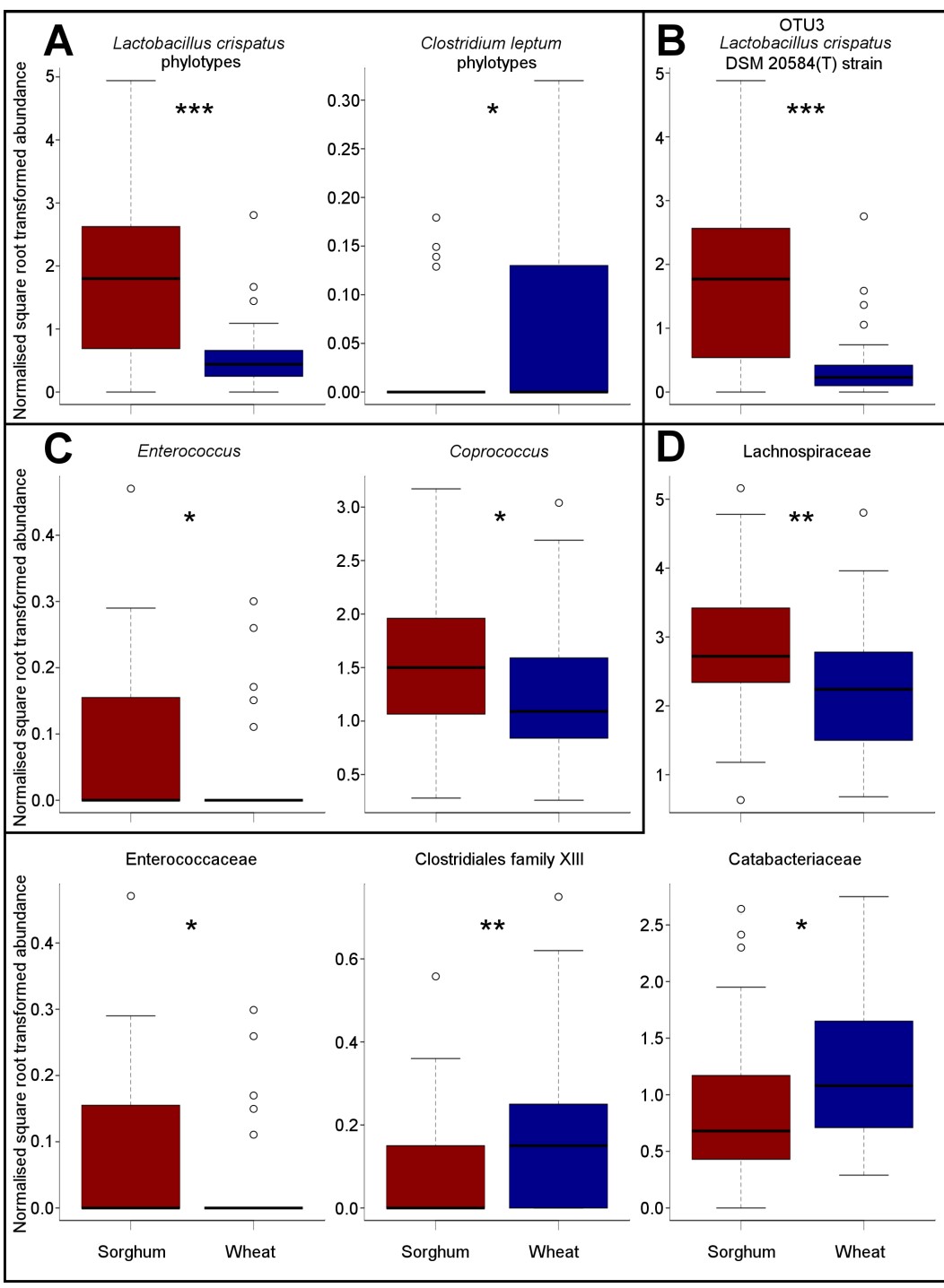

**Figure 4  Caecal microbiota showing significant differences in normalised square root transformed abundance between dietary treatments (indicated by colours).** '(A)', '(C)' and '(D)' show taxa at, respectively, the species, genus and family levels. '(B)' shows the shift in OTU3, which identity was assigned (using EzTaxon) to 100% *L. crispatus* DSM 20584(T) strain. Statistically significant differences are indicated ('\*': $P \leq 0.05$; '\*\*': $P \leq 0.01$; '\*\*\*': $P \leq 0.001$). *L. crispatus* phylotype was assigned to 21 OTUs, all DSM 20584(T) strains, with an averaged similarity (avg. sim.) of 92.67%. *Clostridium leptum* phylotype was assigned to 30 OTUs, all DSM 753(T) strains (avg. sim. = 87.94%).

4C) translated into equal changes at the family level (Enterococcaceae) (Fig. 4D). Also, the Lachnospiraceae family had significantly higher numbers in chickens fed with sorghum than in those fed with wheat ($P = 0.003$). Additionally, two families, Clostridiales family XIII and Catabacteriaceae, showed significantly higher abundances in wheat-fed chickens than in sorghum-fed chickens ($P = 0.008$ and $P = 0.012$, respectively) (Fig. 4D). No other significant shifts were found at any other taxonomic level ($P > 0.05$).

### Associations between microbiota and performance measures

The RDA analysis showed that, at the OTU level in both dietary treatments, gut microbiota composition was significantly associated with AME ($P = 0.007$), but not to FCR ($P = 0.206$), BWG ($P = 0.156$) or FE ($P = 0.061$). Out of the 98 OTUs showing significant shifts between dietary treatments, 54 OTUs (53.02%) showed individual, significant correlations with at least one of the growth performance measures (Table 1). More than 50% of these OTUs ($n = 30$) were significantly correlated with AME values, and all the OTUs showing positive correlations ($n = 11$) were significantly more abundant in sorghum-fed birds than in wheat-fed birds (Table 1). Particularly, four OTUs assigned to a phylotype closest to *L. crispatus* (OTUs 3, 221, 35 and 503) showed consistent positive correlations with AME values. Indeed, *L. crispatus* phylotypes were positively correlated with AME ($\rho = 0.268$, $P = 0.011$). On the contrary, all the OTUs showing negative correlations with AME ($n = 19$) were significantly more abundant in wheat-fed chickens. Most of these OTUs were assigned to either the genus *Clostridium* or the order Clostridiales (OTUs 401, 664, 299, 429, 127, 340, 244, 92, 261, 163, 16, 199, 342 and 148). Nevertheless, some other OTUs within the order Clostridiales were positively correlated with growth performance: nine OTUs assigned to this order showed significant, negative correlations with FCR values (OTUs 401, 127, 92, 261, 409, 550, 355, 682 and 416). Moreover, the family Clostridiaceae was positively correlated with AME values ($\rho = 0.250$, $P = 0.017$). Additionally, eight OTUs (assigned to various taxa) which were significantly prevalent in sorghum-fed birds, showed a significant positive correlation with BWG values (OTUs 123, 511, 162, 43, 54, 192, 321 and 9). At higher taxonomic levels, *Lactobacillus helveticus* was significantly negatively correlated with BWG ($\rho = -0.210$, $P = 0.047$). No other taxa at any other taxonomic level showed significant correlations with any of the growth performance measures.

## DISCUSSION

### Between-diet comparisons of growth performance

Overall, our results indicate that a sorghum-based diet increased AME, BWG and FE, but resulted in poorer FCR of chickens, compared to a wheat-based diet. These findings partially agree with other studies, showing that birds fed on a sorghum-based diet were more efficient in extracting energy from food (high AME values) (*Black et al., 2005*), and had higher BWG (*Liu et al., 2015a*; *Selle et al., 2010*; *Torok et al., 2011*) than those fed with wheat. However, contrary to our results, the latter studies found that sorghum-fed chickens had either better (*Liu et al., 2015a*; *Selle et al., 2010*; *Torok et al., 2011*) or similar feed conversion efficiency (*Black et al., 2005*) than wheat-fed chickens. Caecal microbiota produce short chain fatty acids which provide up to 10% of energy to the bird and

**Table 1 Correlation values based on Spearman's tests between OTUs showing significant shifts between dietary treatments and growth performance measures.** $\rho$ values were ordered from largest to smallest. The significant level of each correlation is indicated. Colours indicate whether a given OTU showed significantly higher mean abundance (normalised square root transformed values) in sorghum-fed birds (red) or wheat-fed birds (blue), and the significant level is indicated as 'P (diet)'. Taxonomic assignment is based on the furthermost default QIIME taxonomy against GreenGenes database.

| OTU no. | Taxon | P (diet) | Sorghum | Wheat | ρ values | | | |
|---|---|---|---|---|---|---|---|---|
| | | | | | AME | FCR | BWG | FE |
| 3 | *Lactobacillus crispatus* | *** | 1.79 | 0.37 | 0.337** | ns | ns | ns |
| 221 | *Lactobacillus crispatus* | ** | 0.11 | 0.02 | 0.332** | ns | ns | ns |
| 305 | Clostridium | *** | 0.13 | 0.03 | 0.331** | ns | ns | ns |
| 35 | *Lactobacillus crispatus* | ** | 0.03 | 0.00 | 0.290** | ns | ns | ns |
| 66 | Clostridium | *** | 0.38 | 0.11 | 0.261* | ns | ns | ns |
| 714 | *Lactobacillus* | ** | 0.23 | 0.05 | 0.251* | ns | ns | ns |
| 318 | Clostridiales | *** | 0.10 | 0.01 | 0.235* | ns | ns | ns |
| 247 | *Ruminococcus* | *** | 0.26 | 0.07 | 0.230* | ns | ns | ns |
| 112 | *Faecalibacterium* | *** | 0.19 | 0.03 | 0.219* | ns | ns | ns |
| 503 | *Lactobacillus crispatus* | * | 0.09 | 0.05 | 0.209* | ns | ns | ns |
| 8 | Clostridiales | ** | 1.29 | 0.48 | 0.207* | ns | ns | ns |
| 401 | Clostridiales Family XIII | ** | 0.06 | 0.14 | −0.208* | −0.233* | ns | ns |
| 664 | Clostridiales | * | 0.02 | 0.11 | −0.210* | ns | ns | ns |
| 299 | Clostridiales | * | 0.00 | 0.02 | −0.211* | ns | ns | ns |
| 245 | Ruminococcaceae | * | 0.02 | 0.07 | −0.212* | ns | ns | ns |
| 429 | Clostridiales | * | 0.02 | 0.12 | −0.213* | ns | ns | ns |
| 127 | Clostridiales | *** | 0.01 | 0.09 | −0.216* | −0.283** | ns | ns |
| 340 | Clostridiales | * | 0.00 | 0.04 | −0.217* | ns | ns | ns |
| 244 | Clostridiales | * | 0.01 | 0.05 | −0.226* | ns | ns | ns |
| 92 | Clostridiales | * | 0.12 | 0.26 | −0.231* | −0.222* | ns | ns |
| 261 | Clostridiales | *** | 0.15 | 0.69 | −0.233* | −0.229* | ns | ns |
| 163 | Clostridiales | ** | 0.27 | 0.66 | −0.235* | ns | ns | ns |
| 16 | Clostridiales | * | 0.05 | 0.22 | −0.239* | ns | ns | ns |
| 199 | Clostridiales | * | 0.03 | 0.20 | −0.247* | ns | ns | ns |
| 202 | *Coprococcus* | *** | 0.05 | 0.22 | −0.257* | −0.242* | ns | ns |
| 589 | Ruminococcaceae | * | 0.00 | 0.02 | −0.265* | −0.211* | ns | ns |
| 77 | Lachnospiraceae | *** | 0.16 | 0.42 | −0.268* | ns | ns | ns |
| 21 | *Faecalibacterium* | ** | 0.67 | 1.16 | −0.278** | ns | ns | ns |
| 342 | Clostridiales | * | 0.00 | 0.03 | −0.315** | ns | ns | ns |
| 148 | Clostridiales | ** | 0.03 | 0.13 | −0.317** | ns | ns | ns |
| 123 | *Clostridium* | *** | 0.17 | 0.02 | ns | 0.378*** | 0.251* | 0.384*** |
| 409 | Clostridiales | ** | 0.00 | 0.04 | ns | −0.208* | ns | ns |
| 550 | Clostridiales | * | 0.00 | 0.03 | ns | −0.219* | ns | ns |
| 48 | Catabacteriaceae | *** | 0.18 | 0.54 | ns | −0.219* | ns | ns |
| 355 | Clostridiales | * | 0.19 | 0.52 | ns | −0.224* | ns | ns |
| 682 | Clostridiales | * | 0.00 | 0.03 | ns | −0.228* | ns | ns |
| 68 | Tenericutes order RF39 | *** | 0.07 | 0.31 | ns | −0.236* | ns | ns |

**Table 1** (*continued*)

| OTU no. | Taxon | P (diet) | Sorghum | Wheat | ρ values | | | |
|---------|-------|----------|---------|-------|-----|-----|-----|-----|
| | | | | | AME | FCR | BWG | FE |
| 416 | Clostridiales | ** | 0.00 | 0.14 | ns | −0.242* | ns | ns |
| 266 | Ruminococcaceae | * | 0.01 | 0.06 | ns | −0.290** | ns | ns |
| 85 | *Ruminococcus* | *** | 0.03 | 0.17 | ns | −0.308** | ns | ns |
| 233 | *Blautia* | * | 0.04 | 0.09 | ns | −0.341** | ns | −0.218* |
| 511 | Clostridiales | * | 1.78 | 1.49 | ns | ns | 0.301** | 0.277** |
| 162 | *Lactobacillus* | * | 0.17 | 0.11 | ns | ns | 0.284** | 0.329** |
| 43 | *Coprococcus* | ** | 0.68 | 0.47 | ns | ns | 0.279** | 0.237* |
| 54 | Lachnospiraceae | * | 0.59 | 0.44 | ns | ns | 0.234* | 0.253* |
| 192 | *Lactobacillus* | * | 0.22 | 0.10 | ns | ns | 0.222* | 0.212* |
| 321 | *Lactobacillus* | * | 0.02 | 0.00 | ns | ns | 0.213* | ns |
| 9 | Clostridiales | ** | 1.88 | 1.33 | ns | ns | 0.207* | 0.270** |
| 203 | Clostridiales | * | 0.03 | 0.10 | ns | ns | −0.226* | −0.249* |
| 724 | *Lactobacillus helveticus* | * | 0.00 | 0.03 | ns | ns | −0.245* | ns |
| 298 | Lachnospiraceae | * | 0.21 | 0.09 | ns | ns | ns | 0.214* |
| 14 | Ruminococcaceae | ** | 0.04 | 0.28 | ns | ns | ns | −0.226* |
| 398 | Tenericutes order RF39 | * | 0.02 | 0.12 | ns | ns | ns | −0.239* |
| 110 | Catabacteriaceae | * | 0.03 | 0.13 | ns | ns | ns | −0.255* |

**Notes.**

ns, not significant.

\*$P \leq 0.05$.

\*\*$P \leq 0.01$.

\*\*\*$P \leq 0.001$.

additionally, caecal contents backflow seed the gut, both up and downstream, via reverse peristalsis or reflux of digesta (*Sonnenburg & Backhed, 2016*). These offer ways in which caecal microbiota are translocated into the intestinal tract thus affecting energy metabolism and performance both directly and indirectly.

Previous research indicated that chickens showing high AME values, tend to show low FCR values because of a low consumption of feed (*Stanley et al., 2016*). However, in our study, birds offered a sorghum-based diet ate more feed compared with those offered a wheat-based diet. These results may suggest a preference for sorghum- vs. wheat-based diets in chickens, in agreement with *Liu et al. (2015a)* and *Liu et al. (2015b)*. Interestingly, only birds fed with a wheat-based diet showed a significant positive correlation between FE and FCR values (i.e., high feed intake leading to poor growth performance, or vice-versa). In either one or the other case, this correlation indicates that birds fed with wheat showed a trade-off between feed consumption and the ability to transform feed into body weight. Such a trade-off did not occur in sorghum-fed chickens, suggesting that their efficiency in converting feed to body mass was not compromised by the amount of feed they ate. Despite this advantage, birds fed on a sorghum-based diet showed comparatively higher FCR values than those fed on a wheat-based diet, indicating that energy in sorghum was used less efficiently than the energy in wheat. The issue of sorghum-based diets in the Australian poultry industry has been extensively reviewed (see *Liu et al. (2015b)* and references within), and some studies have linked sorghum-based diets to inconsistent or

suboptimal broiler performance. These associations might have been produced by anti-nutritive factors present in sorghum (*Selle, Liu & Cowieson, 2013*). Particularly, kafirin concentrations in Australian sorghum cultivars might have increased as an accidental consequence of breeding programs. Also, these cultivars contain phenolic compounds (*Liu et al., 2015b*). These factors could have contributed to the lower utilisation of energy from sorghum-based diet relative to that from wheat-based diets found in this study. Despite this potential disadvantage, *Liu et al. (2015b)* encouraged to rectify these issues by using white sorghums with low kafirin levels. Our results support these findings based on the positive effects that this diet had on AME and BWG values.

A number of studies have suggested a range of supplements can be added to sorghum-based diets in order to improve productivity. For instance, reductions in FCR performance of cereal-fed chickens could be improved by adding distillers dried grains with solubles (DDGS) (*Jacobs & Parsons, 2013*). Also, the latter study indicated that DDGS might, under certain conditions, have additional benefits by increasing AME, caecal microbiota, and gizzard size. Alternatively, *Selle et al. (2016)* showed that adding sodium metabisulphite and protease in sorghum-based diets enhanced starch utilisation, ultimately improving bird growth performance by decreasing FCR and increasing AME and BWG. These supplements could improve the feed conversion efficiency in sorghum-fed chickens.

## Diet-induced changes in caecal microbiota and associations with growth performance

Diet has been recognised as a major modulator of intestinal microbiota. This conclusion has been mainly drawn from microbiota comparisons following the application of diets that are fundamentally very different, such as high fat vs. low fat or, even with more extreme differences, in plant-based vs. Western diet (*Sonnenburg & Backhed, 2016*). In these cases the differences in microbiota are just as extreme as the differences between the diets. Here we compared two similar grain-based diets and although we did find some differences, they were not large as the differences in dietary compositions were not large either. Between-diet comparisons at the microbiota community level showed differences in composition, but not in diversity. Differences in composition were only detected when using Unweighted Unifrac Distance (presence-absence). These results suggest that differences in diet produced a subtle yet significant impact in the structure of the caecal microbial community, confirmed by between-diet shifts in abundance of less than 19% of all OTUs. This is in contrast with *Torok et al. (2011)*, who reported significant differences in microbiota composition in the ileum, but not in the caecum of wheat-fed chickens and sorghum-fed chickens. Despite the low effect of diet on caecal microbiota composition found in our study, a number of caecal bacteria showed strong and contrasting correlations with the growth performance of the birds. Particularly, our results show that the abundance of 30 OTUs were correlated with AME values. These results were supported by the finding that, at the community level, microbiota composition was associated with AME values. Our results are consistent with previous studies, since *Torok et al. (2011)* and *Stanley et al. (2012)* found associations between the composition of caecal microbiota and avian growth performance, at least in wheat-based diets.

Most of the microbiota in both dietary treatments belonged to the orders Lactobacillales and Clostridiales, in agreement with other studies (*Torok et al., 2011*). Particularly, high numbers of phylotypes assigned to *Lactobacillus crispatus* were recorded, yet these were much higher in sorghum-fed birds than in wheat-fed birds, indicating that the abundance of this species is partly influenced by diet-related differences. However, a study found that chickens fed with sorghum showed lower numbers of lactobacilli than chickens fed with wheat (*Shakouri et al., 2009*), in disagreement with our results. Generally, *L. crispatus* is considered beneficial because it produces hydrogen peroxide, tolerates gastric juice and bile salts, and has high surface hydrophobicity (*Mota et al., 2006*). This taxon, along with other strains, is involved in the reduction of pathogen loads, such as *Salmonella* species (*Van der Wielen et al., 2002*). Despite it being generally recognised as a beneficial bacterium, its associations with chicken performance have been mixed. *Stanley et al. (2016)* found that the abundance of *L. crispatus* was associated with high bird performance, since a number of OTUs assigned to this taxon showed either positive correlations with AME values or negative correlations with FCR values. Moreover, *Mignon-Grasteau et al. (2015)* indicated that *L. crispatus* was associated with growth performance, since they found significantly lower abundance in high-FCR than in low-FCR birds. Contrary to the latter two studies, bacterial phylotypes related to *L. crispatus* along with two other *Lactobacillus* species, were linked to low avian growth performance (*Torok et al., 2011*). Also, *Konsak et al. (2013)* found that an OTU which was most closely related to *L. crispatus*, was more abundant in birds with low AME values. Nevertheless, our results reinforce those previously found by our group (*Stanley et al., 2016*), suggesting that the abundance of *L. crispatus* is an indicator of high productivity, and that certain strains of this species may enhance bird growth performance. Although the mechanisms behind this beneficial effect remain unexplained, we hypothesise that *L. crispatus* could increase energy availability through production of short-chained fatty acids. Our hypothesis is supported by *Meimandipour et al. (2010)*, who showed the butyrogenic effects of lactobacilli strains in a simulated chicken caecum. These findings support the conclusion that some strains/OTUs most closely related to *L. crispatus* could be performance-related bacteria that could be studied as candidates to enhance productivity in poultry.

Additionally, sorghum-fed birds showed high abundance of another Lactobacillales member, the *Enterococcus* genus. This genus encompasses several species, with a few considered beneficial, such as *E. faecium* and *E. faecalis* (*Devriese & Pot, 1995*). Nevertheless, other species of *Enterococcus* found in the gastro-intestinal tract of chickens have been associated with infectious diseases, such as *E. cecorum*, related to osteomyelitis (*Kense & Landman, 2011*). One study associated the increased abundance of the *Enterococcus* genus and the Enterobacteriaceae family in the ileum with low feed conversion efficiency in sorghum-fed chickens (*Lunedo et al., 2014*). In our study, we did not observe any significant correlations between *Enterococcus* and FCR, but further research could clarify whether this genus is involved in the high FCR values observed in sorghum-fed birds.

Several taxa within the order Clostridiales showed different abundances between dietary treatments and contrasting correlations with growth performance measures. Particularly in this group, the Lachnospiraceae family was the most abundant taxon, showing higher

numbers in sorghum- than in wheat-fed chickens. This family is a butyrate-producing group considered as beneficial plant degraders and strong producers of short chain fatty acids (*Biddle et al., 2013*), and its numbers are reduced in gastrointestinal illnesses such as chronic liver cirrhosis (*Chen et al., 2011*) and inflammatory bowel disease (*Berry & Reinisch, 2013*). Lachnospiraceae have been associated with improved growth performance in chickens (low FCR values) (*Stanley et al., 2012*). Furthermore, *Torok et al. (2011)* found 4 OTUs assigned to Lachnospiraceae associated with high performance (low FCR values), and 1 OTU assigned to the same taxon, associated with low performance (high FCR values) in the caecal microbiota of chickens. Our results support the latter two studies, since one OTU assigned to this family was found to be positively correlated with BWG. Additionally, the *Coprococcus* genus was relatively more abundant in sorghum- than in wheat-fed chickens. One OTU assigned to this genus was positively correlated with BWG and FE, but another OTU was negatively correlated with FCR and AME, indicating mixed associations with growth performance. Previously, *Stanley et al. (2016)* found three different OTUs assigned to *Coprococcus* to be positively correlated with FCR, which indicates that this genus might have negative consequences for chicken growth performance.

Also, there were three bacterial taxa within the Clostridiales order showing higher abundance in wheat-fed than in sorghum-fed chickens: the families Catabacteriaceae and Clostridiales family XIII, and the *Clostridium leptum* phylotype. Catabacteriaceae, a relatively unknown family that has been found in polluted water (*Codony et al., 2009*), has been associated with poor feed conversion efficiency in chickens (*Stanley et al., 2016*). Also, although little is known about Clostridiales Family XIII, *Singh et al. (2012)* found Clostridiales Family XIII linked to broilers showing high FCR values. Additionally, *Steelman et al. (2012)* found significantly higher abundance of this taxon in horses with chronic laminitis than in healthy horses. In our study, we did not find any association between the three above-mentioned Clostridiales bacteria and chicken productivity. However, based on what other studies have reported, high abundance of these bacteria may result in poor bird performance. Future research efforts could study how differences in the proportions of cereals in feed influence the survival of these Clostridiales taxa in the chicken gut microbiota. Interestingly, a number of other bacteria within the Clostridiales order showed mixed associations with growth performance measures. Out of several OTUs (identified as either Clostridiales or *Clostridium*) which were prevalent in wheat-fed chickens, some showed negative correlations with AME, whereas others showed negative correlations with FCR. Such inconsistency of patterns might have been produced because Clostridiales is a high polyphyletic order. There seems to be some ambiguity in the taxonomy of several *Clostridium* species (*Yutin & Galperin, 2013*) despite *Biddle et al. (2013)* recommending major revisions. Species within this genus are a mixture of gram positive and negative bacteria, and some are not even anaerobes (*Fåk & Bäckhed, 2012*). Our results, in accordance with *Stanley et al. (2016)*, suggest that some members of the *Clostridium* genus (from the Lachnospiraceae family) could be associated with better performance, whereas a number of other unidentified taxa in this order could be associated with low performance. Thus, in agreement with other studies (*Rinttilä & Apajalahti, 2013*; *Stanley et al., 2016*), the

term clostridia should not be related with poor performance and/or health in chickens and beneficial and detrimental clostridia should be clearly distinguished.

## CONCLUSIONS

In summary, our results suggest that differences in growth performance depending on cereal type used in the diet were correlated with changes in specific members of the microbiota in the caecum of chickens. Also, our study reinforces the hypothesis that specific bacteria could be used as probiotics to improve performance, given the associations of the caecal microbiota and growth performance measures that we found. Future studies could test this hypothesis in order to promote long-term sustainability for the poultry industry.

## ACKNOWLEDGEMENTS

The data was analysed using the Isaac Newton High Performance Computing System at Central Queensland University. We wish to acknowledge the support from Jason Bell in all aspects of high performance computing. We also thank Derek Schultz, Evelyn Daniels and Kylee Swanson (SARDI) for their assistance with animal trials, and Honglei Chen (CSIRO) for operating the Roche 454 sequencer.

### Funding

This research was conducted within the Poultry CRC, established and supported under the Australian Government's Cooperative Research Centres Program under project CRC 2.1.5. DS is ARC DECRA Fellow. The funders had no role in study design, data collection and analysis, decision to publish, or preparation of the manuscript.

### Grant Disclosures

The following grant information was disclosed by the authors:
Australian Government's Cooperative Research Centres Program.

### Competing Interests

The authors declare there are no competing interests.

### Author Contributions

- Eduardo Crisol-Martínez analyzed the data, wrote the paper, prepared figures and/or tables, reviewed drafts of the paper.
- Dragana Stanley and Robert J. Moore conceived and designed the experiments, analyzed the data, contributed reagents/materials/analysis tools, reviewed drafts of the paper.
- Mark S. Geier and Robert J. Hughes conceived and designed the experiments, performed the experiments, contributed reagents/materials/analysis tools, reviewed drafts of the paper.

## Animal Ethics

The following information was supplied relating to ethical approvals (i.e., approving body and any reference numbers):

The Animal Ethics Committees of the University of Adelaide (Approval No.S-2011-218) and the Department of Primary Industries and Resources, South Australia (Approval No. 25/11) approved this study. All animal work was conducted in accordance with the national and international guidelines for animal welfare.

## Data Availability

The complete dataset is available on the MG-RAST database (http://metagenomics.anl. gov/mgmain.html?mgpage=project&project=mgp20398) under library ID mgl538406.

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
