# Peer review of "Sorghum and wheat differentially affect caecal microbiota and associated performance characteristics of meat chickens"

_PeerJ, doi:10.7717/peerj.3071_

## Round 0.1 · original submission · Minor Revisions

As you will see, both expert reviewers agreed that your manuscript makes a valuable contribution to the field. Nevertheless, especially reviewer 1 provided several relevant suggestions for further improvement of especially the discussion, and indicated some issues with data analysis. Furthermore, in your introduction you mention that feed has a major impact on the microbiota composition, but from your study that is not obvious. This fact could be addressed more explicitly in the discussion.

·

Basic reporting

In general, the manuscript is well written and adheres to Peer J policies.

Experimental design

The DNA extraction should be explained in a bit more detail, i.e. the reader should be aware that "raw extracts" were obtained by bead beating and ammonium acetate precipitation with subsequenct purification on Qiagen stool columns.

Validity of the findings

A comment on the correlation analysis: the Pearsons correlation analysis is more suitable normally distributed data. This may be the case for performance, but probably not for microbiota data. I suggest to use the Spearman correlation analysis instead, which is more robust against non-normally distributed data.
Discussion
L156ff This paragraph discusses performance, but not microbiota, thus the title should be changed. The paragraph could also be shortened to focus on the main point of interest (microbiota impact on performance)
L310f The discussion of lactobacilli (together with clostridia) as dominating bacteria in the hind gut needs a little revision. The cited reference (Dumonceaux et al. 2006) states that lactobacillis dominated the small-, but not the large intestine (68% Clostridiales vs. 25% Lactobacilliales). This is consistent with other studies that show the Clostridiales are dominating the hind gut of poultry.
L329ff I am missing discussion of a possible the mode of action for the beneficial L. crispatus. It is of interest that this species may be an indicator, but why would it interfere with AME? Unfortunately, bacterial metabolite data is missing, so a possible increased uptake of SCFA as energy source can not be ascertained. Nevertheless, do the authors think that the species itself is responsible or is it just an indicator, while the truly responsible bacteria may be of different origin than lactobacilli?
L350 please add animal species that were studied in the references
L393 If there is no indication on the physiology of the bacteria found to impact performance, they may just be indicators with no actual effect. Thus, the conclusion that "probiotic bacteria" were found is not justified.

Additional comments

In general, I have one major comment on the discussion of the data: The bacteria that were found to correlate with performance may only be indicators without any effect on performance. There should be some arguments related to their interaction with the host. Digestion of energy delivering nutrients occurs exclusively in the small intestine and thus, the analysis of the hind gut may not reflect the situation in the small intestine. How could bacteria influence metabolizable energy? Directly via energy delivering metabolites, indirectly via less energy costs for maintenance/ detoxification? Discussion of these points may enhance the presented data.

Reviewer 2 ·

Basic reporting

The article is well written with sufficient introduction and background

Experimental design

The research question is to compare the effects of a wheat diet and a sorghum diet on the performance and caecal microbiota composition of broiler chickens. It is not clear why the timeframe (day 15 to day 25) was taken.
Fig 4 C1 (Enterococcus) and D2 (Enterococcaceae) look exactely the same, is this correct?

Validity of the findings

no comments

---

## Round 0.2 · accepted · Accept

Thank you for carefully revising the manuscript, which has much improved.

·

Basic reporting

no comment

Experimental design

no comment

Validity of the findings

no comment

Additional comments

no comment